# Residual Stand Damage under Different Harvesting Methods and Mitigation Strategies

**Anil Raj Kizha [1,*]** , **Evan Nahor [2]** , **Noah Coogen [3]** , **Libin T. Louis [1]** and **Alex K. George [1]**

1   School of Forest Resources, University of Maine, Orono, ME 04469, USA; libin.thaikkattil@maine.edu (L.T.L.); alex.george@maine.edu (A.K.G.)

2   Thomas Donnelly Logging, New York, NY 12847, USA; evan.nahor@maine.edu

3   LandVest Timberland, Jackman, ME 04945, USA; ncoogan@landvest.com

*   Correspondence: anil.kizha@maine.edu; Tel.: +1-(207)-581-2851

**Abstract:** A major component of sustainable forest management are the stands left behind after the logging operation. Large mechanized harvesting equipment involved in current forest management can inflict damage on residual trees; and can pose a risk of mortality from diseases, natural calamities, and/or degrade future economic value. The primary objective of this study was to evaluate the residual stand damage under different harvesting methods and silvicultural prescriptions i.e., crop tree release (CTR), diameter limit cut (DLC), and overstory removal (OSR). The second objective was to evaluate the intensity and frequency of damage occurring on the bole, canopy, and root at tree and stand level. The third objective was to document strategies adopted globally to minimize stand damage due to timber harvesting. Five harvest blocks implementing three silvicultural prescriptions, were selected as the treatments across two different industrial timberlands in central and northern Maine (Study Site (SS) I and II, respectively). A hybrid cut-to-length (Hyb CTL) and whole-tree (WT) harvesting method were employed for conducting the harvest in SS I and II, respectively. Systematic transect sampling was employed to collect information on type, frequency, and intensity of damages. The inventory captured 41 and 8 damaged trees per hectare with 62 and 22 damages per hectare from SS I and SS II respectively. Bole damage was the most frequent damage across all treatments. The Hyb CTL had lower damage density (damage per ha) and severity compared to WT. The average number of trees damaged per ha was higher for CTR prescriptions compared to DLC. There were no significant differences in the height of the damages from the ground level between treatments within each study site; however, there was a significant difference between the study sites. Species damaged was directly related to the residual trees left behind and was dominated by American beech, yellow birch, sugar maple, and eastern hemlock. Finally, the study provides strategies that can be adopted at different forest managerial phases to mitigate residual stand damage.

**Keywords:** forest operations; forest management; partial harvest; silvicultural prescription; timber harvesting; tree injury

## 1. Introduction

In the Northeastern United States, partial harvests compose the majority of timber harvest, moreover, in the state of Maine, 92% of the total harvests performed in 2017 were partial harvest [1]. This can be attributed primarily to the silviculture of most commercial species in the region, including northern hardwoods such as red maple (*Acer rubrum* L.) and yellow birch (*Betula alleghaniensis* Britt.), along with conifers like eastern white pine (*Pinus strobus* L.) and spruce (*Picea* spp.), which are established and grow best naturally under partial shade [2]. Thinning operations are also key to manage overstocked stands and to restore ecosystem services [3–5]. A major component of these partial timber harvests are the trees left behind. With large mechanized harvesting equipment involved, the damage is unavoidable and can be inflicted on the crown, bole (stem), or roots of the residual trees.

These damages can potentially expose the trees to pathogens and other pests, which can weaken tree or even lead to tree mortality [6–8]. Even if parasites do not get into the tree, the loss of bark could result in defect that may retard tree growth. Wounds to the stem can also disrupt physiological functions such as nutrient and water uptake, loss of potential secondary growth and can lead to wood discoloration [9]. Overall, residual stand damage can pose severe challenge to sustainable forest management, as damaged trees would barely be able to perform its ecological functions.

The typical objective for commercial timberland management is to improve the wood-lot characteristics, such as growing larger trees, improving species composition, and promoting regeneration. The residual trees may be intended for future growth in economic value, wildlife habitat promotion, seed supply, general aesthetics, or the ecological and cultural value of certain trees to landowners. Residual stand damage is a detriment to any of these goals, as some of the trees intended to be kept, are damaged during the harvest. Tavankar et al. [10] reported that 12–23% of residual trees from a harvested site were damaged in a selection cutting operation. Tree damages are usually centered around areas where extraction and felling occurs [4,11].

Residual stand damage is affected by several factors, including site characteristics, species composition, stand stocking, silvicultural prescription, harvest season, machine selection, and operator proficiency. Although any operation with heavy harvesting equipment results in residual stand damage, machine selection plays a major role in the amount, type and intensity of residual stand damage [12]. Often based on the silvicultural objectives, cut-to-length (CTL) operations are the preferred harvesting method due to their ability in minimizing residual stand damage [13,14]. A modified version of CTL, "hybrid cut-to-length" (Hyb CTL) is typical to the northeastern forests of the United States, where a feller-buncher aids in the directional felling of the trees followed by the harvester/dangle-head processor processing them within the stand. Such harvesting methods are widely believed to ensure slash on the trail (armor for roots and soil), exhibit increased productivity compared to that of conventional CTL, and have less adverse impact on the regeneration. However, the total cost of harvest is generally higher for Hyb CTL compared to a conventional CTL due to the additional equipment (feller-buncher) [5].

Damage to residual trees could impact long-term management goals as the healing of wounds are dependent on several factors including species, age, slope, elevation, and wound type; furthermore, healing rates can range between 10–19 mm yr$^{-1}$ [15]. The relationship between tree damage and future loss in revenue has not been well documented [9] and the only available information at large comes from the last century. While estimating the net log volume (scaling), tree wounds are generally accounted for as a deduction factor. In North America, decay from logging damage constituted for an annual loss of 0.75% of the gross volume for western hemlock [8,16]. Studies have reported that losses in financial revenue can add up to 7–20% at the time of final harvest, and in one case the value of wounded spruce logs were reduced by 30% compared to sound logs of similar size [8,17]. As most frequent damage occurs at the lower part of the tree, which is also the most valuable log having the largest diameter and volume, residual tree damage could degrade future economic value of the stand [18–20]. A potential scar, or damage to the first log on a tree can turn a valuable veneer or select log into pulpwood, thereby diminishing the value of the entire tree [8,15].

The objective of this study was to evaluate residual stand damage in three different silvicultural prescriptions, i.e., crop tree release (CTR), diameter limit cut (DLC), and overstory removal (OSR) employing a hybrid cut-to-length (Hyb CTL) and a conventional whole-tree (WT) harvesting operations. Specific objectives were to: (1) compare residual stand damage between the different harvesting methods and silvicultural prescriptions; (2) evaluate the intensity and frequency of damages occurring on the bole, canopy, and root, both at tree and stand level; (3) document strategies adopted globally to minimize residual stand damage due to forest management activities.

## 2. Materials and Methods

### *2.1. Site Description*

Two industrial timberlands were selected for the study (Figure 1).

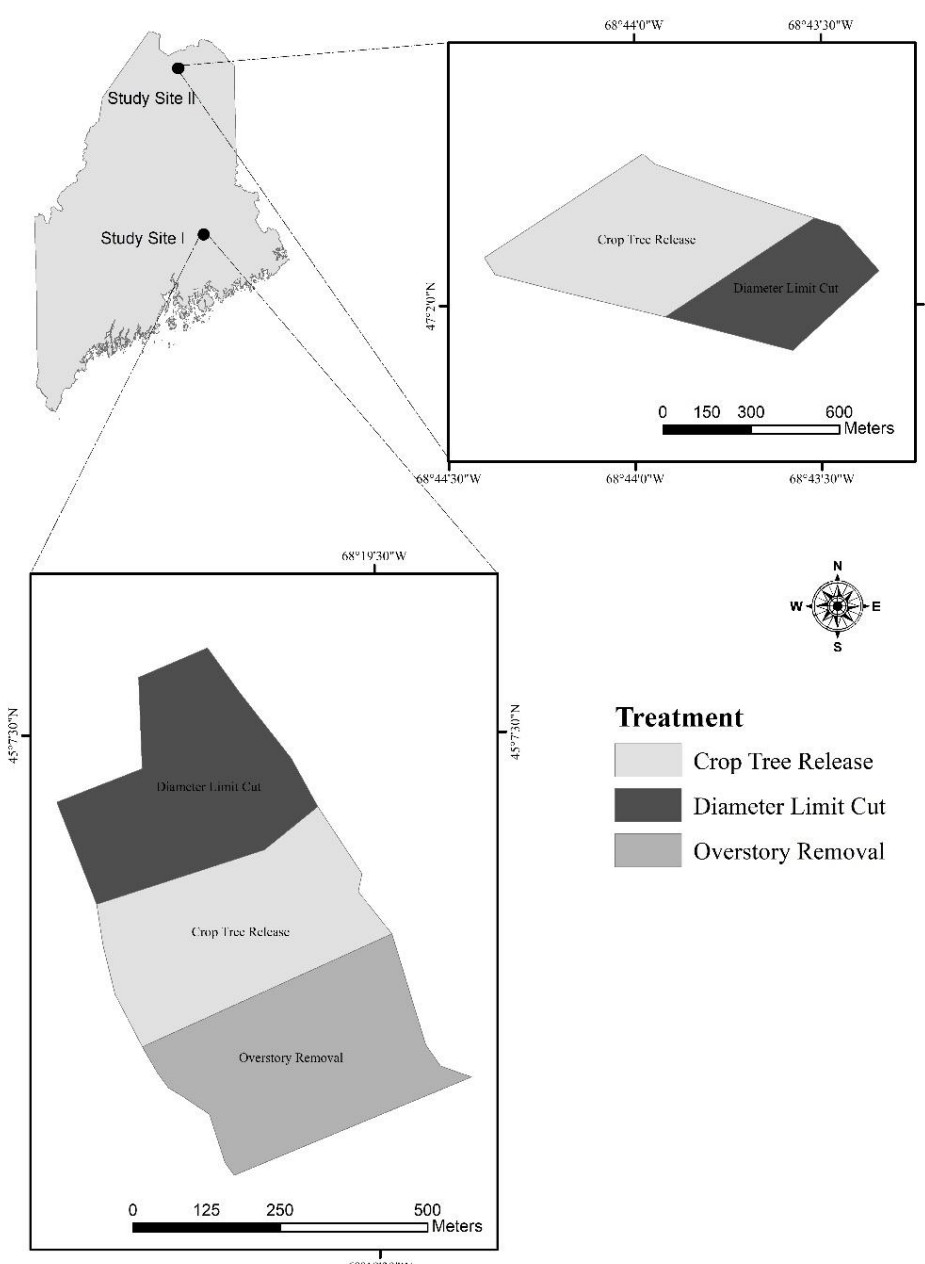

**Figure 1.** Locations of Study Sites I and II with treatment units in each site to evaluate the residual stand damage in varying harvesting methods and silvicultural prescriptions.

2.1.1. Study Site I (SS I)

The harvest site was located in Grand Falls township in Penobscot County, central Maine, USA (45°70′17.39″ N and 68°19′47.35″ W) and was managed by American Forest Management. The three timber harvest blocks selected were next to each other. The total area of the harvested blocks was 31.97 ha. Average annual rainfall and snowfall was 115 and 218 cm, respectively. The high temperature averaged 20 °C and 4 °C for summers and winters, respectively [21]. Soil types were predominantly Howland silt loam and Monarda-Burnham complex. These soils were glacial in origin, gravelly silt loams, poorly

drained with depth to hardpan of 30–50 cm [22]. The slope of the site was uniform and relatively flat (<9%).

SS I was a mixedwood forest (hardwood and softwood) with the dominant species being eastern hemlock (*Tsuga canadensis* (L.) Carr.) and yellow birch (*Betula alleghaniensis* Britt.). Other species included balsam fir (*Abies balsamea* (L.) Mill.), aspen (*Populus* spp.), American beech (*Fagus grandifolia* Ehrh.), red spruce (*Picea rubens* Sarg.), white birch (*Betula papyrifera* Marshall), and eastern white pine (*Pinus strobus* L.). The stand was last harvested in the early 1980′s during the last spruce budworm outbreak in Maine with chainsaws and cable skidder.

### 2.1.2. Study Site II (SS II)

SS II was located 10 km west of the town of Eagle Lake, Northern Maine, ME, USA (47°2′7.54″ N and 68°44′10.95″ W, Figure 1) and was owned and managed by J.D. Irving Woodlands LLC. The harvested blocks were 48.50 ha. Average annual rain and snowfall was 99 and 246 cm, respectively. The average temperature for the year was 3 °C (ranging between −20–24 °C) [21]. The three soil types present were Chesuncook-Elliotsville, Monson-Elliotsville, and Ragmuff-Monson, all having similarities in terms of profile characteristics, slope effect, soil texture, and coarse material. All the soil series range from moderately well drained to fairly excessively drained and the slope (approximately 12%) was consistent throughout the study site [23].

Unlike SS I, SS II was primarily a northern hardwood stand, with a mix of some softwoods, such as red spruce, white spruce (*Picea glauca* (Moench) Voss), balsam fir, eastern white pine, and northern white-cedar (*Thuja occidentalis* L.).

### 2.2. Stand Inventory

At 10% sampling intensity, 20 Basal Area Factor (BAF) variable radius plots were set in a systematic design for conducting pre- and post-harvest inventory. All trees above 8 cm in diameter at breast height (DBH, 1.37 m from the ground) were measured and the parameters recorded included species, DBH, and geographical coordinates for the plot center. Tree heights were measured for every sixth tree of a given species. Dead snags and diseased trees were recorded separately. The inventoried trees were also examined for any deformities such as rot, twists, broken tops, and holes. Basal area per hectare ($m^2 \ ha^{-1}$) and stand density (trees $ha^{-1}$) were calculated for each study site.

### 2.3. Silvicultural Prescriptions

Both study sites were subjected to two versions of partial harvesting—crop tree release (CTR) and diameter limit cut (DLC); SS I had a third prescription—overstory removal (OSR). Each contrasting prescription implemented denoted extremes in market conditions and landowner objectives. In general, CTR retained healthy and straight trees with the intention of enhancing growing conditions for the residual stand. On the other hand, DLC harvested all high-value trees (from which a sawlog could be obtained) and left poorer quality stems as the residual stand. OSR resembled a clear-cut, however retained trees of 13 cm or less in DBH as the residual stand.

For all prescriptions, damaged and defective trees were removed during the harvest. Snags, boundary, and cavity trees were retained.

### 2.3.1. Study Site I

Removal intensity for all prescriptions, except OSR, were similar and was to harvest half of the basal area (BA) or up to 15 $m^2 \ ha^{-1}$. The following were the harvest priority for individual trees, high to low: (1) damaged and defective trees except for cavity trees; (2) fir and aspen over 15 cm DBH; (3) American beech over 12 cm DBH, except for smooth-barked trees: (4) spruce over 31 cm DBH and small-crowned trees; (5) White and yellow birch over 25 cm DBH; (6) white pine over 46 cm DBH; (7) hardwoods over 13 cm DBH and poorly-formed trees; (8) eastern hemlock over 13 cm and poorly-formed trees.

OSR harvested all trees above 13 cm of DBH. The residual stand comprised of trees which were predominantly smooth-barked beech, oak, snags, and cavity trees kept for wildlife purposes; along with large, non-veneer quality yellow birch trees were retained every 60 m apart as seed trees.

### 2.3.2. Study Site II

SS II was split up into two treatment blocks with CTR II and DLC II (Figure 1). The harvest removed 50–70% of the basal area or leaving 7–14 $m^2 \cdot ha^{-1}$. Species and quality to be removed with the respective priority ranking from high to low: (1) all merchantable balsam fir and spruce over 20 cm DBH; (2) northern white cedar over 15 cm DBH; (3) all merchantable American beech (except if it not infected with the beech bark disease); (4) merchantable white birch and poplar; (5) mixed hardwoods damaged or above 41 cm. In terms of residuals, current and future sawlogs were left across all diameter classes for CTR II. It was also expected that they were not at risk and would remain in the stand for the next 15–20 years.

### 2.4. Harvesting Operation

The harvests were conducted during the summer of 2017 and 2018, for SS I and II respectively, with the intention of minimizing site disturbance and allowing some soil exposure. The same machines and operators were employed for all prescription treatments within each study site. A conventional ground-based whole tree (WT) method was employed in SS I and the operation lasted for two months. A feller-buncher (John Deere 753 G) was utilized for felling and bunching. Extraction was carried with two grapple skidders (Caterpillar 625 G); delimbing and topping was carried out by a stroke delimber (John Deere 200 LC) at the landing. A slasher was used for merchandising as well as loading trucks [22].

For SS II, a hybrid cut-to-length (Hyb CTL) harvesting method was used. A feller-buncher (John Deere 853, John Deere, Moline, IL, USA) cut the trees and left them in the trails for a dangle-head processor (John Deere 753, John Deere, Moline, IL, USA) to delimb and process into specified log length within the unit. A forwarder (John Deere 1910, John Deere, Moline, IL, USA) then carried the logs to the roadside to be picked up by self-loading trucks. The operation was carried out during July through October of 2018 [24]. The total amount of wood harvested from each treatment block was obtained from the scale tickets.

### 2.5. Data Collection

Residual stand damage data were collected a month after respective harvests and lasted for 2–3 weeks. Multiple sampling techniques were considered, including systematic plot, random plot, blocks along skid trail, and systematic transect. All of these techniques generated similar range of accuracy, although some methods had certain comparative advantage over the others [11]. Systematic transect sampling was chosen as it allowed trail variability and was best suited for the relative shape of the treatment blocks (rectangular; Figure 1). The transects also provided a more accurate representation of harvested and unharvested areas within the stand [11].

Systematic transects (9.14 m wide) were laid perpendicular to the skid trails. The first transect was randomly selected, followed by laying down subsequent transects approximately 80 m apart from each other. There were five and four transects for SS I and II, respectively. A major limitation of systematic sampling is associated with establishing and staying on the centerline of the transect during data collection [11]. However, this obstacle was overcome by using GPS units and tablets with the transects already mapped, making the entire process comparatively smoother and faster. Additionally, this also ensured that the data collected were inside the transect itself. Avenza maps was the software used to display the map with a real-time GPS location. For boundary line trees, perpendicular distance from the transect center to the center of the trees along the vertical axis at the DBH were measured.

Only damaged trees were accounted for the residual stand damage survey. The measurements taken were broadly divided into damage and tree-level information. The damage-level information constituted of (1) type of damage; (2) size of damage; (3) severity of damage; and (4) vertical location (height) of the damage on the tree from the ground; and the tree-level information were (1) number of trees damaged; (2) number of damages per tree; (3) DBH and species; and (4) GPS location of the damaged tree. Type of damage was broadly divided into damage to canopy, bole, and root damage. Wound size (surface area) was calculated as a product of the maximum length and width of the exposed wood [9]. Canopy injury was measured conceptually in terms of percentage of the canopy removed based on the undamaged tree form.

To be considered for damage, the minimum threshold was a wound area above 4.62 cm$^2$ (bigger than the size of US¢ 25; adopted from the regional forest management industrial standards). All damages for bole and root were assigned a severity index (SI) of 1–10, to assess the intensity of injury [4]. Severity index were based on visual observations (Figure 2) and were classified as: SI = 1 (negligible injury, damage area above 4.62 cm$^2$); SI = 3 (light injury, damage area of 120 cm$^2$ with the bark being removed and minimal cambium damage); SI = 5 (medium injury, damage area of 500 cm$^2$ and cambium has been cut into); and SI = 10 (heavy injury with an entire portion of the cambium has been removed (>3000 cm$^2$).

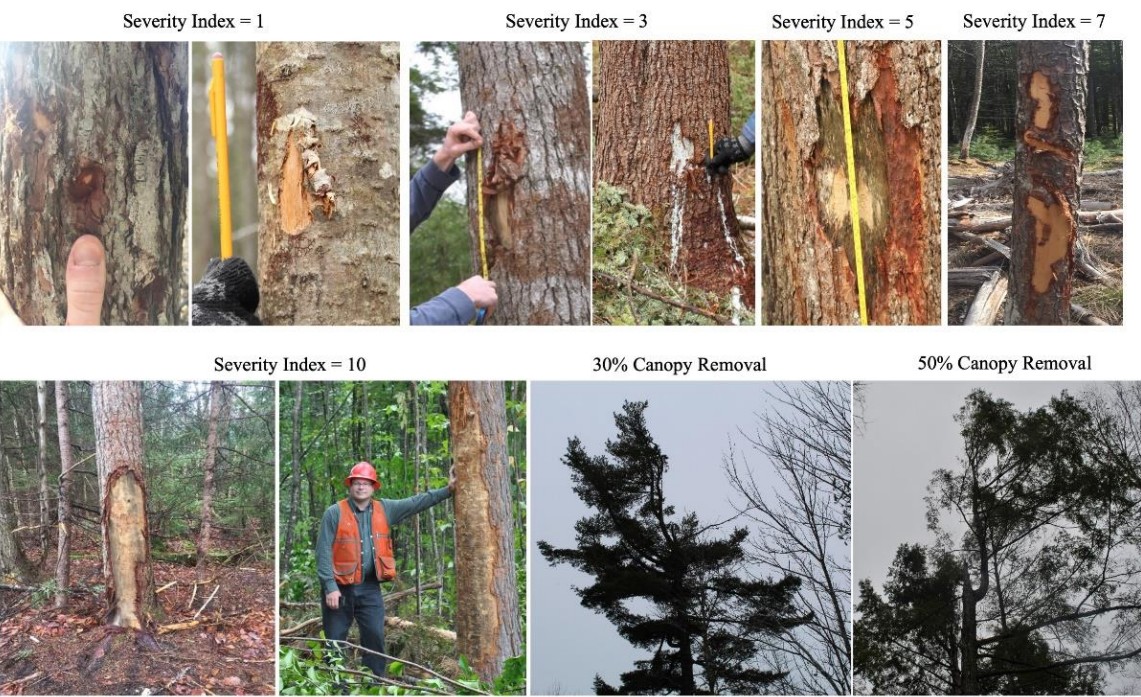

**Figure 2.** Illustrations of severity index (SI) and damages incurred on the tree due to timber harvesting operation. SI were based of visual observation.

For understanding the damages from a perspective of the prescription, the transect-level information was normalized to stand level by dividing the number of damages and trees damaged with the total inventoried area for respective treatment blocks.

### 2.6. Statistical Analysis

Analysis of Variance (ANOVA) were performed using R statistical software (version 3.6.1) to determine if a significant difference ($p < 0.05$) existed between treatments, and study sites for the following: (1) DBH class (pre-harvest); (2) severity index; (3) size of damage; (4) number of damages per trees; (5) vertical position of damage on the trees. Tukey test was done to determine significant difference ($p < 0.05$) between individual

treatments. Additionally, ANOVA was done to evaluate if significant difference existed between the transects for each treatment. Normal distribution and homogeneity of variance were verified prior to ANOVA using Shapiro-Wilk and Levene test, respectively.

### 2.7. Literature Review

For accomplishing the third objective, a literature search was conducted using major online databases and library catalogs: CrossRef, Scopus, Google Scholar, and Web of Science. The initial search started with "residual stand damage" as the keyword, which yielded 26 scientific articles. After careful analysis of those articles, three more synonyms, "tree wound", "tree injury", and "wood damage" were used to gather more literature. Additionally, the reference section of the previously selected articles was also utilized for more specific search. A total of 52 scientific articles (peer reviewed and non-peer reviewed) were identified as relevant. The strategies discussed in each article were later categorized into pre-, during, and post-harvest.

## 3. Results and Discussion

### 3.1. Stand Conditions

A total of 5.91 and 18.12 ha was inventoried for SS I and II respectively, which comprised 18 and 37% of the total harvested area (Table 1). There was no significant difference in DBH between the treatment blocks for SS I ($p = 0.295$) prior to the harvest; however, there was a significant difference within SS II ($p = 0.027$).

**Table 1.** General characteristics of the treatment blocks for the residual stand damage study inventoried using variable radius plots with 20 BAF (Basal Area Factor) prism. Stand inventory summarized all standing trees above diameter at breast height of 8 cm.

| | | Study Site I | | | Study Site II | |
|---|---|---|---|---|---|---|
| | | **CTR I** | **DLC I** | **OSR I** | **CTR II** | **DLC II** |
| Treatment area (ha) | | 10.52 | 10.12 | 11.33 | 32.31 | 16.19 |
| Residual stand damage transects (ha) | | 1.87 | 2.03 | 2.01 | 11.81 | 6.31 |
| Basal area (m$^2$·ha$^{-1}$) | Pre-harvest | 22 | 37 | 45 | 29 | 30 |
| | Post-harvest | 19 | 18 | 8 | 7 | 8 |
| Stand density (trees ha$^{-1}$) | Pre-harvest | 687 | 1108 | 2710 | 1028 | 1578 |
| | Post-harvest | 482 | 187 | 210 | 517 | 85 |
| Total wood harvested (ton) | | 1031 | 1151 | 1913 | 2646 | 1664 |
| Skid trail length (m) | Total | 2608 | 2458 | 1668 | 6567 | 4838 |
| | Average | 318 | 254 | 408 | 1009 | 873 |

Where, DLC—Diameter Limit Cut; CTR—Crop Tree Release; OSR—Overstory Removal.

### 3.2. Residual Stand Damage

A total of 62 and 22 damages per hectare were recorded from SS I and II, respectively. The residual stand damage reported for all treatments were acceptable in the land managing companies' standards. ANOVA for all parameters (DBH, number of damages per trees, SI, size of wound, and vertical position of damage) between the transects within each treatment blocks were not significant (for all except one), ratifying that the transects were true representatives of the treatment blocks (Table 2). Validation using ANOVA is a general approach for testing bias within the population that involves comparing two different samples (transects) drawn from the same population [25].

**Table 2.** *P*-value from the ANOVA conducted to validate data collected from the transects within each treatment block. The values indicated in the parenthesis are test statistics and degree of freedom.

| | Study Site I | | | Study Site II | |
| --- | --- | --- | --- | --- | --- |
| | **DLC I** | **CTR I** | **OSR I** | **DLC II** | **CTR II** |
| Area of injury (cm$^2$) | 0.474 (0.519, 51) | 0.770 (0.086, 76) | 0.137 (2.232, 151) | 0.640 (0.220, 144) | 0.653 (0.202, 216) |
| Bole damage severity (1–10) | 0.971 (0.001, 49) | 0.123 (2.447, 60) | 0.232 (1.443, 121) | 0.205 (1.62, 144) | 0.073 (3.252, 208) |
| Height from base (m) | 0.696 (0.154, 48) | 0.054 (3.897, 58) | 0.103 (2.696, 121) | 0.397 (0.723, 144) | 0.110 (2.580, 207) |
| Number of damage. tree$^{-1}$ | 0.764 (0.091, 44) | 0.456 (0.562, 63) | 0.992 (0.062, 129) | 0.023 * (5.533, 50) | 0.478 (0.507, 94) |

Where, DLC—Diameter Limit Cut; CTR—Crop Tree Release; OSR—Overstory Removal. * There was a significant difference between the transects within the treatment.

### 3.2.1. Damage Type

In SS I, there were 10, 11, and 42 damages per hectare to canopy, root, and bole, respectively. Whereas in SS II, one damage per hectare each to canopy and root; and 19 damages per hectare to the bole. Among the 240 sampled trees that were damaged from SS I, 128 were damaged in the bole alone, 26 in roots alone, and 17 in the canopy alone. For 27 trees, damages were recorded in the bole and roots, 32 trees for damages in the bole and the canopy, three trees for damages in roots and the canopy, and seven trees were observed for all three damage types. Out of the 148 trees damaged in SS II, 109 were damaged in the bole alone, 18 were damaged in the bole and roots, 12 were damaged in the bole and in the canopy, six had all three damage types, and three in just the canopy. The most common damage type was injury to the bole, which comprised 67 and 89% of the total number of damages in each site (SS I and II), respectively (Table 3 and Figure 3).

**Table 3.** Tree damages (total and mean) recorded for the various treatment blocks in Study Site I and II. Values in parenthesis indicates standard error.

| | Study Site I | | | Study Site II | |
| --- | --- | --- | --- | --- | --- |
| | **DLC I** | **CTR I** | **OSR I** | **DLC II** | **CTR II** |
| **Total** | | | | | |
| Bole damage | 50 | 65 | 133 | 146 | 201 |
| Root damage | 18 | 13 | 31 | 3 | 20 |
| Canopy damage | 12 | 24 | 22 | 3 | 19 |
| **Mean** | | | | | |
| DBH of damaged trees (cm) | 24 (±1.12) | 19 (±1.59) | 9 (±0.31) | 11 (±0.89) | 24 (±1.46) |
| Height from base (m) | 0.76 (±0.008) | 0.76 (±0.007) | 0.64 (±0.004) | 0.85 (±0.003) | 0.85 (±0.004) |
| Area of injury (cm$^2$) | 1430 (±3.01) | 529 (±0.75) | 298 (±0.56) | 159 (±29.95) | 356 (±51.51) |
| Length (cm) | 67 (±0.77) | 58 (±0.66) | 49 (±0.39) | 27 (±2.45) | 30 (±2.46) |
| Width (cm) | 21 (±0.15) | 9 (±0.08) | 6 (±0.03) | 6 (±0.31) | 12 (±0.71)) |
| Bole damage severity (1–10) | 3 (±0.36) | 2 (±0.31) | 3 (±0.21) | 2 (±0.15) | 2 (±0.15) |
| Root damage severity (1–10) | 1 | 3 | 4 | 2 | 2 |
| Canopy damage height from base (m) | 7 | 7 | 5 | 10 | 11 |
| Percentage of canopy removed (%) | 10 | 30 | 78 | 37 | 15 |

Where, DLC—Diameter Limit Cut; CTR—Crop Tree Release; and OSR—Overstory Removal.

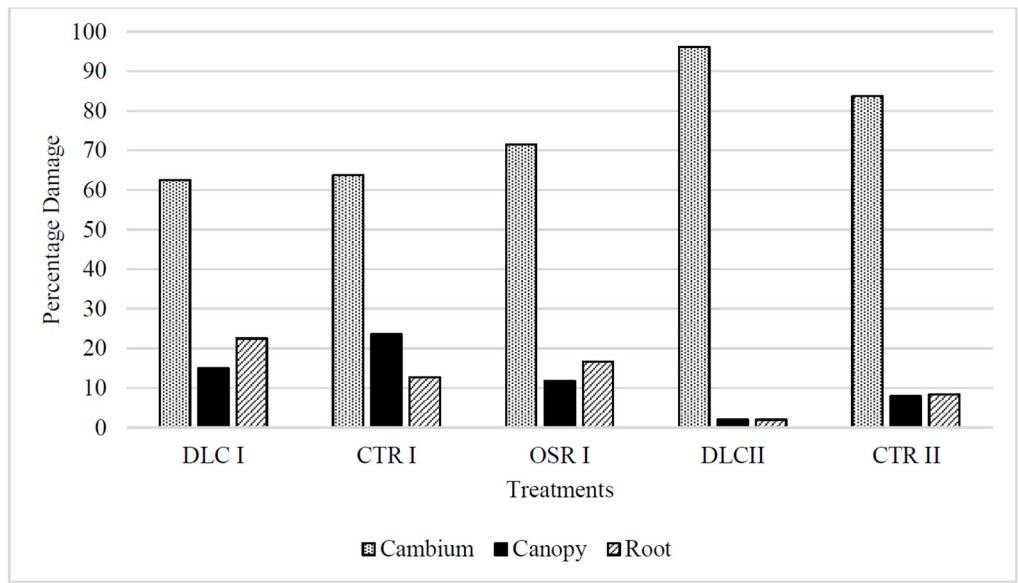

**Figure 3.** Types of residual stand damage recorded from treatment blocks using systematic transect sampling method. Where DLC—Diameter Limit Cut; CTR—Crop Tree Release; OSR—Overstory Removal.

During extraction (particularly for skidding in SS I) on trail turnings, the log bunch had a tendency to rub on standing trees, causing bole damage in the trails. Most of the damage, especially root and bole damage, were concentrated around the trails. The width of the skid trails ranged between 3.5–4.2 m for both study sites and was wider as it approached the landing. Previous studies have reported highest amount of damages occur in the course of extraction [26].

Attributes for root damage can be equipment running over during both felling and extraction. While SS I had 10.49 incidences of root damage per hectare, SS II accounted for 1.33 (Table 3). This can be directly attributed to the slash armored trails in the Hyb CTL stands, which protected the root system. Additionally, the wood was carried in Hyb CTL, rather than being dragged as with the WT stands. The average root severity index was also less for SS II (SI = 3) compared to SS I (SI = 5) (Figure 4). Depending on the severity of root damage, most often structural integrity of the tree is compromised, with the trees have a higher probability to be wind thrown during the next storm event. This was particularly observed on trees having a shallow root system. Severity index values along with the damage size were directly related to probability of the tree to be infected by pathogens [27,28]. A SI of one (between 5 and 10 cm$^2$ or a minor bark injury) could be considered inconsequential due to low risk level of the tree getting infected by wood-infecting fungi [29]. As the wound size increases above 10 cm$^2$, the risk of decay occurrence is directly proportional to the size of injury and SI [27,28]. Nyland [30] reported a wound size of 975 cm$^2$ have a 50% probability of being infected by decay within 10 years.

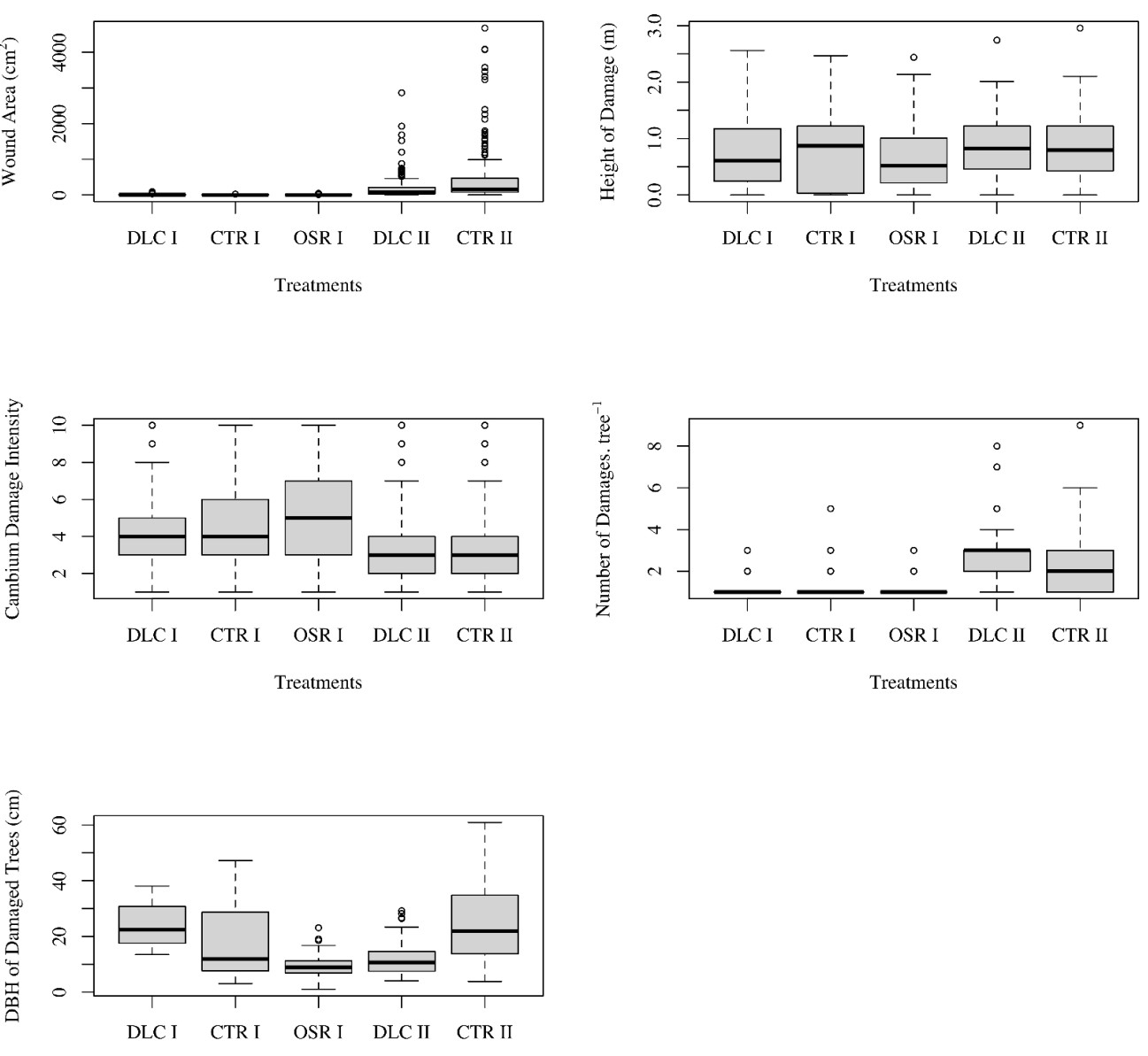

**Figure 4.** Boxplot for parameters collected from treatment blocks in the residual stand damage data. Where DLC—Diameter Limit Cut; CTR—Crop Tree Release; OSR—Overstory Removal; "I" and "II" refers to study site classification.

There were 9.81 and 1.50 canopy damages per hectare recorded for SS I and II, respectively. However, if the OSR I treatment was deducted from the total value for SS I, the number of canopy damage dropped to 9.23 damages per hectare. Canopy damages were detected close to felled trees in both study sites. During felling, the crowns can hit each other and break branches, creating canopy damage [10,31,32]. Canopy damage negatively affects the stand productivity and growth rate of trees.

### 3.2.2. Impacts due to Harvesting Methods and Silvicultural Prescriptions

The total number of damages, trees damaged, and SI for root damage were highest for OSR I (93 damage ha$^{-1}$, 65 trees ha$^{-1}$, and 8, respectively; Table 4 and Figure 4). This can be attributed to the small sized trees (less than 13 cm) left as the residual stand. The average DBH for a damaged tree being 9 cm for OSR I was a direct result of this (Table 3).

**Table 4.** Tree damages normalized for stand-level for the various treatment blocks on a per hectare basis. The number of damages and trees damaged obtained from the transects were divided with the total inventoried area for respective treatment blocks.

|  | Study Site I | | | Study Site II | |
|---|---|---|---|---|---|
|  | DLC I | CTR I | OSR I | DLCII | CTR II |
| Total number of trees damaged | 47 | 65 | 131 | 52 | 95 |
| Total number of wounds | 80 | 102 | 186 | 152 | 240 |
| Average number of wound. tree$^{-1}$ | 1.70 ($\pm$0.07) | 1.57 ($\pm$0.08) | 1.42 ($\pm$0.04) | 2.92 ($\pm$0.22) | 2.53 ($\pm$0.15) |
| Transect area (ha) | 2.03 | 1.87 | 2.01 | 6.31 | 11.81 |
| Wood harvested (tons. ha$^{-1}$) [a] | 113.74 | 98.00 | 168.84 | 102.78 | 81.89 |
| Number of wounds. ha$^{-1}$ | 39 | 55 | 93 | 24 | 20 |
| Trees damaged. ha$^{-1}$ | 23 | 35 | 65 | 6 | 12 |

Where, DLC—Diameter Limit Cut; CTR—Crop Tree Release; and OSR—Overstory Removal. [a] The total wood harvested from each block obtained from the scale tickets were divided with the area of treatment block.

A significant difference was present in damage size between all treatments from both study sites ($p < 0.01$, Tables 3 and 5). This could be potentially due to the difference in intensity of harvest in each treatment block, species composition, and tree size of the residual stand. Another potential source can be the experience level and the ecological awareness of the machine operators (between study sites), which was not examined for this study [4].

**Table 5.** Statistical values for the ANOVA performed between the treatments for measured parameters. $P < 0.01$ shows a statistical significance.

|  | *p*-Value | F Statistics | F Value | Degrees of Freedom |
|---|---|---|---|---|
| DBH of damaged trees | <0.001 | 2.395 | 49.969 | 4 |
| Height from base | <0.010 | 2.387 | 3.585 | 4 |
| Area of injury | <0.001 | 2.387 | 22.205 | 4 |
| Average number of wounds. tree$^{-1}$ | <0.001 | 2.395 | 37.654 | 4 |
| Bole damage severity | <0.001 | 2.387 | 17.027 | 4 |

In SS I, the average DBH of damaged trees were larger in DLC I compared to CTR I; and a reverse trend was observed for SS II (Table 3). Tukey test results showed that there was a significant difference between treatments for both study sites ($p < 0.001$, Table 6). Similarly, there were no clear patterns exhibited for the wound size and severity index in terms of silvicultural prescriptions implemented; however, there was a significant difference between the treatment blocks ($p < 0.001$; Table 5). The reason could be due to the previous mentioned attributes. For example, in SS II, CTR II had larger wounds on an average compared to DLC II ($p < 0.001$; Table 6); a reverse trend was observed for SS I and Tukey test showed no significant difference. In terms of the severity index, SS I recorded more intense damage for DLC I compared to CTR I, and both being equal for SS II. Even though Tukey test showed a significant difference between study sites, there was none between treatments within the study site (Table 6). The size and intensity of the wound along with the location plays a major role in determining how fast decay spreads from the wound [7,33]. It is worth noting that a large number of bumper trees were left on DLC I which might have contributed to the high average wound size (Table 3). However, results from the Tukey test showed no significant difference (Table 6). Bumper trees are used on sharp skid trail turnings to streamline bunches during extraction and/or to protect desired residual trees from bole and root damage. Bumper trees are sacrificial, low value trees which usually are afflicted with high amount of damage due to extraction (generally SI between 7–10); therefore, cut at the end of the harvest or left for wildlife purposes.

**Table 6.** *P*-value from the Tukey test conducted to further analyze statistical difference among the different treatments. Q-statistics provided in the parenthesis.

| | CTR I | OSR I | DLC II | CTR II |
|---|---|---|---|---|
| **Area of Injury** | | | | |
| DLC I | 0.9 (0.03) | 0.9 (0.15) | 0.001 * (8.81) | 0.05 * (3.87) |
| CTR I | | 0.9 (0.20) | 0.001 * (11.19) | 0.006 * (4.81) |
| OSR I | | | 0.001 * (7.90) | 0.115 (3.40) |
| DLC II | | | | 0.001 * (6.37) |
| **Bole damage severity** | | | | |
| DLC I | 0.9 (0.67) | 0.463 (2.34) | 0.013 * (4.50) | 0.005 * (4.94) |
| CTR I | | 0.126 (3.35) | 0.038 * (4.02) | 0.014 * (4.49) |
| OSR I | | | 0.001 * (9.23) | 0.001 * (10.30) |
| DLC II | | | | 0.9 (0.38) |
| **Average number of wounds per tree** | | | | |
| DLC I | 0.9 (0.02) | 0.9 (0.33) | 0.001 * (10.90) | 0.001 * (8.15) |
| CTR I | | 0.9 (0.35) | 0.001 * (11.88) | 0.001 * (9.12) |
| OSR I | | | 0.001 * (13.79) | 0.001 * (11.28) |
| DLC II | | | | 0.020 * (4.33) |
| **Height from base** | | | | |
| DLC I | 0.9 (0.02) | 0.739 (0.66) | 0.745 (1.65) | 0.695 (1.77) |
| CTR I | | 0.694 (1.77) | 0.721 (1.71) | 0.666 (1.84) |
| OSR I | | | 0.016 * (4.42) | 0.006 * (4.83) |
| DLC II | | | | 0.9 (0.08) |
| **Diameter at breast height (DBH) of damaged trees** | | | | |
| DLC I | 0.031 * (4.12) | 0.001 * (12.69) | 0.001 * (8.54) | 0.9 (1.22) |
| CTR I | | 0.001 * (9.11) | 0.004 * (5.02) | 0.001 * (6.31) |
| OSR I | | | 0.301 (2.74) | 0.001 * (17.85) |
| DLC II | | | | 0.001 * (11.32) |

* shows significant difference.

The average number of trees damaged per ha was higher for CTR prescriptions compared to DLC (Table 4). Leaving more residual trees and the density of the residual stand directly correlates with the amount of residual damage. Since there were more trees left in the CTR more trees were damaged. ANOVA showed a significant difference between the treatments in terms of damages per tree ($p < 0.001$, Table 5). Further analysis using Tukey test revealed no significant difference existed between treatments within the SSI; however, there was a significant difference in SSII ($p = 0.02$, Table 6). A significant difference was also observed between the study sites. Silvicultural prescriptions can influence the extent of stand damage, based on several factors but not limited to the basal area removed as well as the stand conditions. Nyland [30] reported that there was a direct relation between the tree removal intensity and damage occurrence. Machine movement within the stand for picking specific trees of higher rank in the pecking order, could result in increased damage for selection prescriptions [15]. Additionally, the amount of wood harvested is directly proportional to the number of machine passes (during extraction) which is further proportional to the likelihood of tree damage due to machine movement. A different trend was observed for SS I, with CTR I having more wound ha$^{-1}$ and the lowest removal intensity compared to the other treatment units within the study site.

Results from ANOVA showed a significant difference in the height of wound between the two study sites ($p < 0.007$; Table 5); however, Tukey test did not reveal any difference (Table 6). Additionally, there was no significant difference between different treatments within a study site ($p = 0.36$ and 0.95 for SS I and II, respectively; Tables 5 and 6), which could be attributed to the same harvesting machines that operated within each study site. Bole damages (62%) were mainly located in the lower parts of the tree, within 1 m above

the ground. Further, 24% of the total bole damage were located between 1 and 1.5 m above the ground. Bole damages were higher from the ground for the CTL (SS II, average 0.85 m), compared to the WT (SS I, average 0.76 m) (Table 2), which can be attributed to the in-wood processing in the former. Other studies also reported similar results [3,4,12,34–37]. Further the wounds away from the ground tends to reduce the detrimental impacts on the damaged tree and can be attributed to reduction in the probability of pest or pathogen infection, thereby increasing the wound healing rate [7,15,38].

In terms of the severity of bole damage, a significant difference was observed ($p < 0.001$) between study sites (i.e., harvesting methods); however, similar to height of damage no significant difference was observed between treatments within the study site ($p = 0.06$ and 0. 77 for SS I and II, respectively; Tables 5 and 6). In general, the wound size and damage severity (for both bole and roots) in SS I (WT method) was larger compared to SS II (Hyb CTL; Figure 5). Cudzik et al. [12] obtained similar results comparing a traditional CTL to a tree-length method. This can be directly related to the payload being dragged (skidded) in SS I compared to being carried (forwarded) in SS II. Additionally, CTL method had fewer machine passes compared to the WT [26,39]. Interestingly, Han and Kellogg [18] reported majority of the residual damage was done in the felling and processing phase by the harvester (67%) compared to the extraction phase (29%) in a CTL harvesting method.

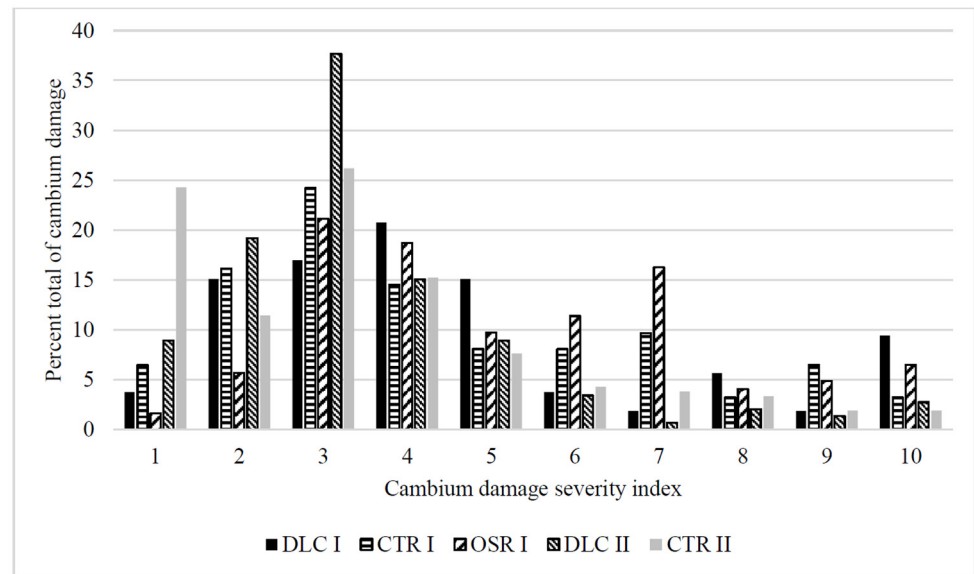

**Figure 5.** Bole damage intensity shown by severity index for the treatment blocks. Where DLC—Diameter Limit Cut; CTR—Crop Tree Release; OSR—Overstory Removal; "I" and "II" refers to study site classification.

### 3.2.3. Species Composition and Diameter Classes of Damaged Trees

Yellow birch and eastern hemlock constituted around 69% of the total trees damaged in SS I. For SS II, sugar maple, yellow birch, and American beech were the species most damaged. Amongst which, sugar maple and yellow birch were the desired crop trees for the CTR II treatment and constituted most of the residuals left behind. Furthermore, in SS II, American beech, accounted a large portion of small diameter trees in the stand prior to harvest with a basal area of only 5.35 $m^2$ $ha^{-1}$ but 791 trees $ha^{-1}$. One of the requirements of the CTR prescriptions were to remove poor quality stems and was specifically targeting the American beech (mostly affected with the beech bark disease), and yellow birch. In the DLC, since removing poor quality stems was not the intent of the harvest, most of the species were left behind. Since they were sprouts, they were in dense patches, and were prone to damage by felled trees and harvesting equipment. Another reason for the high damage rate among yellow birch and American beech can be due to the species being found in close proximity to extraction trails.

Resistance and sensitivity to tree wounds varies between species [7,12]. Certain hardwood species (like oak and cherry) have superior decay resistant traits, and converts the dead sapwood (primary metabolites and sugar) to secondary compounds, which eventually helps in checking the spread of pathogen and the wood decay process [9,40]. This portion of the wood tends to have dark color heartwood and are assigned greater value in the market. On the contrary, stained heartwood of other merchantable hardwoods (such as maple, birch, and sweetgum) can reduce value of the wood [9]. Jourgholami and Abari [41] reported a significant relation between species and log losses from tree damage (both in terms of volume and value).

Tavankar et al. [15] reported species along with the stand canopy closure, wound type, and age of the tree had a significant effect on the wound healing rate (0.64–2.40 cm $yr^{-1}$). Furthermore, the fast-growing nature of the tree species also affected wound healing rates. In general, the wound healing rates ranged from 10–50 years depending on the species, wound type, and geographical location [10,42–44]. Smith et al. [45] reported there was 88% chance of smaller wounds (<650 $cm^2$, SI of 6 and below) to be healed within 10 years; however, this dropped to 19% for larger wounds.

Out of the total damaged trees, most trees had a DBH below 15 cm (Figure 6). Prior to harvest, majority of the trees were in diameter size class between 23 to 43 cm. There was a significant difference in the DBH of the damaged between the study sites and treatments within each study site ($p < 0.001$, Tables 5 and 6). Similar to the reasoning for the damages within species composition, the silvicultural prescriptions likely skewed the damage occurrences towards the smaller diameter classes. Ezzati and Najafi [38] also reported similar results with 49% of the total damage inflicting DBH class of <35 cm. Jourgholami and Abari [41] reported a significant relation between DBH class of the damaged trees and tree wounds. However, no conclusion can be made here, as this was dependent on the stand conditions and silvicultural prescriptions. Even though diameter class of the trees is directly related to the radial growth increment, no significant factors have been reported in terms of predicting wound closures [10,46].

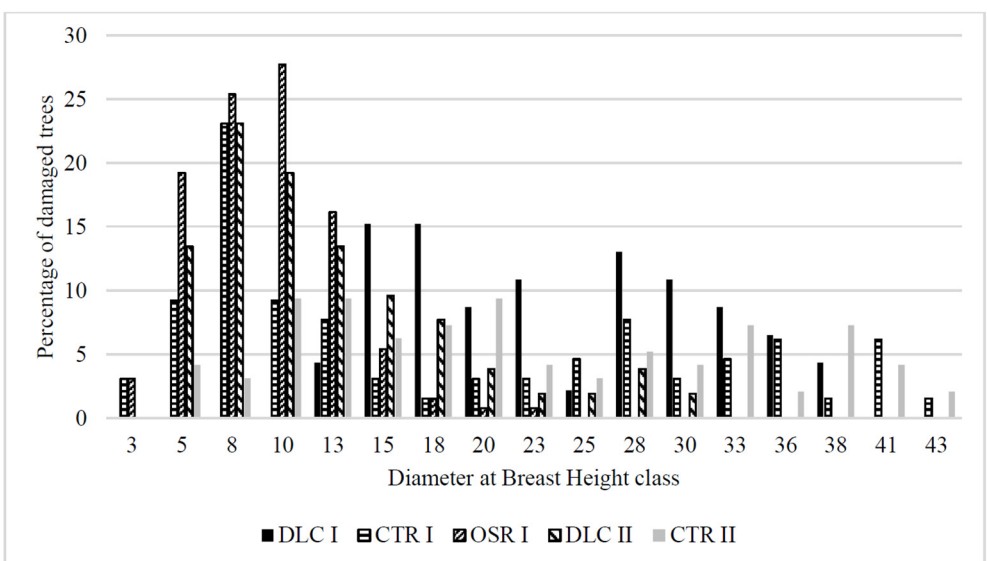

**Figure 6.** Percentage of damaged trees per two-centimeter diameter class Where DLC—Diameter Limit Cut; CTR—Crop Tree Release; OSR—Overstory Removal; "I" and "II" refers to study site classification.

### 3.3. Strategies Adopted for Minimizing Residual Stand Damage

Residual stand damage is an inevitable part of modern mechanized logging. However, occurrences of these damages can be minimized with appropriate precautionary strategies put in place during different stages of forest management. Table 7 provides strategies discussed in different scientific studies conducted globally over the past 20 years. The

strategies are broadly divided into action taken: (1) pre-harvesting; (2) during harvesting; and (3) post-harvesting operation. Most of these strategies are not universal and depends on species and stand composition, silvicultural and managerial objectives, equipment and crew availability, geographical location, and market conditions.

**Table 7.** A literature review on strategies adopted for minimizing residual stand damage in mechanized harvest operations.

| | Strategies | Sources |
|---|---|---|
| | **I. Prior to harvest** | |
| | **Logging crew** | |
| 1 | Gain trust of operators | [15,18,47] |
| 2 | Appropriate training and supervision | [8,15,26,36,38,48–50] |
| 3 | Increase awareness on consequences of mechanical damage | [26,30,51,52] |
| 4 | Experience level of operators and proficiency | [8,18,26,38,48,53] |
| 5 | Incentive programs promoting minimal amount of residual stand damage | [18] |
| | **Silviculture** | |
| 1 | Minimize the number of stand entries over time | [52] |
| 2 | Use crop tree selection methods rather than wide-spread thinning techniques; thereby limiting harvesting activity to limited areas | [38,52,54] |
| 3 | Shorter rotations to minimize decay loss after thinning | [18,55] |
| | **Harvest layout and management** | |
| 1 | Proper harvest layout and skid trail planning | [8,10,18,26,36,38,47,53–56] |
| 2 | Visibly identify/mark the crop trees and use extra caution while working near them. | [18,36,37,47,52,53,57] |
| 3 | Know the site and stand characteristics | [47,52] |
| 4 | Establish maximum acceptable levels of damage (species wise) in the management plan | [55] |
| | **Extraction trail** | |
| 1 | Optimal trail spacing | [18,26,54] |
| 2 | Wider extraction trails | [51] |
| 3 | Straight skid trails | [18,26,49] |
| 4 | Use bumper trees | [47,49,53] |
| 5 | Leaving low stumps on trails | [18,48] |
| 6 | Use old extraction trails | [18,49] |
| 7 | Armor extraction trails with slash | [20,22,58,59] |
| 8 | Remove wood from skid trails first | [47] |
| | **Equipment selection** | |
| 1 | Match equipment type and size to stand and site conditions | [18,26,47,49,52,53] |
| 2 | Balance between productivity and quality of thinning operation | [51] |
| 3 | Choose machines having lesser footprint | [51,53] |
| | **Harvest methods/system** | |
| 1 | Using Cut-to-length method (or extracting smaller sized logs) | [12,27,34,37,47,56,60,61] |
| | **Season** | |
| 1 | Winter harvesting | [8,9,13,18,27,48,51–53,55,59] |
| 2 | Avoid harvest during wet conditions and periods of sap flow | [9,18,47,48,52,53] |
| | **II. During operation** | |
| 1 | Prioritize efforts to reduce injuries to roots/soil first; upper bole/crown next | [18,52] |
| 2 | Recognize that sapling/pole stages are most vulnerable to damage | [9,52,57] |
| 3 | Directional felling | [18,37,38,47,49,53,54] |
| 4 | Extra care to be taken around larger residual trees | [55] |
| 5 | Tree protection device (tree pads) | [18,38,55,60] |
| | **III. Post-harvest** | |
| 1 | Chemical wound decay control | [8] |
| 2 | Regular assessment of stand for decay | [53,55] |

The most discussed strategies within the literature review included: (1) proper harvest layout and skid trail planning; (2) appropriate training and supervision; (3) directional felling; (4) opting CTL method (or bucking into smaller sized logs prior to extraction); and (5) winter harvesting. There were also a few contradictory statements. For example: Vasiliauskas [8] reported application of chemical treatments for tree wounds as potential strategy; whereas Zeglen [47], described wound dressing as unnecessary and can further induce more decay.

Most of the strategies discussed regarding the logging crew are being done at the field level (Table 7); training session through vocational program and loggers' association are common in the region. Strategies in terms of silvicultural prescriptions are dependent on several factors including the landowners' objectives and site conditions. Wider extraction trail can provide more space for the operators to work efficiently thereby minimizing residual stand damage. On the contrary, this option would leave more area under extraction trails leading to reduction in the land for growing trees. The straightness of the skid trails is subjected to site conditions. In Maine, with major portion of forested landscape under water-submerged conditions, straightness of the skid trails is often compromised [13]. Using of extraction trail from previous harvest is also subjected to state of the trail itself. Some of the previous skid trails might have been transformed to streams over the years, in which case should be avoided.

Winter season is when most of the harvest are carried out in the region; however due to rapidly warming winter season, the operational timeframe is becoming shorter [62]. Additionally, during peak season, selection of appropriate equipment, and operators is often challenging due to the multiple harvest being conducted. In such instances, foresters usually prioritize suitable logger for the appropriate stands based on several factor including equipment, prior experience, and market conditions.

## 4. Limitations of the Study

There were several limitations in the study of which the first could be attributed to study sites differing in stand density (pre-harvest) and species composition. As the intensity of harvest, pecking order and harvesting methods employed, were different for both study sites, a direct comparison was not possible in terms of silvicultural prescriptions. Additionally, OSR I had the highest removal intensity compared to other treatments in SS I, which might have skewed the damage parameters discussed.

Another drawback was in terms of the sampling intensity which was 18 and 37% for SS I and II respectively. This was because field data collection had to be completed prior to a major snowstorm prediction for SS I. The smaller sampling percentage in the SS I could account for lesser variability captured. Again, these results would have varied if the harvesting season changed, as summer harvest usually have higher damage rates compared to winter [27]. In addition, the number of the undamaged trees within the transects were not collected; due to which the percentage of total tree damaged could not be evaluated.

Finally, the experience level of the crew operating the harvesting machine (especially during the extraction phase) varied between the study sites. SS I operators had comparatively lesser experience (3 year) than the SS II crew (more than 15 years). Previous studies have shown that residual tree damage is highly correlated to operational techniques and working pattern which is by large determined by the operator's experience [8,18,31,56].

## 5. Conclusions

Quantifying residual damage is crucial in evaluating the success of the harvesting operation in terms of sustainable forest management [26]. Many factors can impact the amount and level of damage during timber harvesting, making scientific research results not universal [12]. In this study, bole wounds were the major form of damage, with majority being located less than a meter from the ground. Most of the damage were attributed to the extraction phase. In terms of silvicultural prescription, the average number of trees damaged per ha was higher for CTR prescriptions compared to DLC. The harvesting method had a profound influence on the damages, with the Hyb CTL inflicting lesser damage to residual trees compared to WT. This comparison between harvesting methods can justify how important it is to select the right equipment based on the harvest stand conditions.

**Author Contributions:** Conceptualization A.R.K.; methodology, A.R.K., E.N., N.C., L.T.L. and A.K.G.; software, L.T.L., and A.K.G.; validation, A.R.K., E.N., N.C., L.T.L., and A.K.G.; formal analysis, E.N., N.C., L.T.L., and A.K.G.; investigation, A.R.K., E.N., and L.T.L.; resources, A.R.K.; data curation, E.N., N.C., L.T.L., and A.K.G.; writing—original draft preparation, A.R.K., E.N., and N.C.; writing—review and editing, A.R.K., E.N., N.C., L.T.L. and A.K.G.; visualization, A.R.K., L.T.L. and A.K.G.; supervision, A.R.K.; project administration, A.R.K.; funding acquisition, A.R.K. All authors have read and agreed to the published version of the manuscript.

**Funding:** This work was supported by U.S. Department of Agriculture/Agricultural Research Service, Forest Products Research Project #5407527; U.S. Department of Agriculture/National Institute of Food and Agriculture (McIntire-Stennis project) through the Maine Agricultural and Forest Experiment Station under Grant number #ME0-41909; and the Cooperative Forestry Re-search Unit (CFRU). Maine Agricultural and Forest Experiment Publication Number 3829.

**Institutional Review Board Statement:** Not applicable.

**Informed Consent Statement:** Not applicable.

**Data Availability Statement:** The data shall be made available upon request.

**Acknowledgments:** We would like to express our gratitude to Noel Lienert for assisting in various aspects of editing and data processing. We would also like to thank Brian Edward Roth, Stephen Dunham, and Jenna Zukswert, CFRU for the support and help in various stages of the study. Our appreciation goes to all the foresters, contractors and machine operators associated with American Forest Management and Irving Woodland LLC for their involvement in the operational aspect of the study.

**Conflicts of Interest:** The authors declare no conflict of interest. The funders had no role in the design of the study; in the collection, analyses, or interpretation of data; in the writing of the manuscript, or in the decision to publish the results.

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
