# Peer review of "Residual Stand Damage under Different Harvesting Methods and Mitigation Strategies"

_sustainability, doi:10.3390/su13147641_

Round 1
Reviewer 1 Report
The design and development of the research experiments are fine, and the authors deserve the publication of their work.
The conclusions are supported by the results, although I miss a relative expression of the number of damaged trees, i.e. not only the comparison of the raw per hectare number of damaged trees but also the percentage of remnant trees that were damaged for each treatment/site.
If the authors added and analyzed these parameter, I think that their study would gain significance
Author Response
Response to Review for “Residual Stand Damage Under Different Harvesting Methods and Mitigation Strategies”
Dated June 7th 2021
Reviewer I
The design and development of the research experiments are fine, and the authors deserve the publication of their work.
The authors would like to express their appreciation for the comment.
The conclusions are supported by the results, although I miss a relative expression of the number of damaged trees, i.e. not only the comparison of the raw per hectare number of damaged trees but also the percentage of remnant trees that were damaged for each treatment/site. If the authors added and analyzed these parameter, I think that their study would gain significance
Data on the residual trees that was not damaged within the transect were not collected. This has been mentioned in Page 5 Line 216. The authors acknowledge this limitation of the study and has further mentioned it in the “Limitation section” in the current version too. Page 18, Line 519-21.
The authors would like to thank the reviewer for the positive comments and critical review. This has helped us in significantly improving the quality and readability of the article.

Reviewer 2 Report
The topic is relevant to the field of interest spanned by the journal readers, and the paper is original. The article has a novelty character due to the way in which it analyses the frequency with which standing trees are damaged when using different harvesting systems. Considering the importance of limiting the damage brought to standing trees, this paper could be of general and international interest. The work methodology is adequate and presented in detail so that it can be reproduced. Both the research planning and the number of samples (and trees/sample) are clearly stated. The paper contains a high amount of data and adequate statistical interpretation that have allowed authors to obtain representative results. The results are appropriately stated and commented. The language is clear.
However, the paper could be further improved in some points, namely:
1- The reference section contains a high number of bibliographic references representative of the topic approached in the paper. In the introduction these references are used as background for the presentation and justification of the topic approached and later on they are used to support the results obtained and the discussions. However, in the paper the authors do not clearly state whether the damage level is high or low, acceptable or not. Also, I feel that the discussion would benefit from providing some terms of comparison for the level of damage corresponding to traditional operations (chainsaw and skidder that have been applied in the past in the studied stands). Otherwise, the readers are informed about the damage level derived from mechanized operations but are left wondering how the new mechanized technology compares with traditional logging technology;
2 – It is not enough to take into consideration only the area (the size) of the wound in order to establish its severity (see the suggestions made in the text, line 223);
3 – There are some inaccuracies in the data regarding the number of damaged trees (lines 280-289) that need to be corrected. It is recommended to check the data in the text and tables (for example Table 6);
4 - In section - 3.3. Strategies adopted for minimizing residual stand damage - this part represents a general strategy, based on the studied bibliography. It would be interesting to come up with a strategy (for minimizing residual stand damage) developed on the basis of the results obtained and applicable to the stands studied under the given conditions and which could be reproduced in other stands located under similar conditions.
Author Response
Response to Review for “Residual Stand Damage Under Different Harvesting Methods and Mitigation Strategies”
Dated June 7th 2021
Reviewer II
For the ease of answering some of the reviewer’s comments, the authors have taken the liberty to divide the comments into further sub-sections.
The topic is relevant to the field of interest spanned by the journal readers, and the paper is original. The article has a novelty character due to the way in which it analyses the frequency with which standing trees are damaged when using different harvesting systems. Considering the importance of limiting the damage brought to standing trees, this paper could be of general and international interest. The work methodology is adequate and presented in detail so that it can be reproduced. Both the research planning and the number of samples (and trees/sample) are clearly stated. The paper contains a high amount of data and adequate statistical interpretation that have allowed authors to obtain representative results. The results are appropriately stated and commented. The language is clear.
The authors would like to express their appreciation for all the positive comments, especially the recognition the reviewer has provided regarding the importance and uniqueness of this study.
However, the paper could be further improved in some points, namely:
- The reference section contains a high number of bibliographic references representative of the topic approached in the paper. In the introduction these references are used as background for the presentation and justification of the topic approached and later on they are used to support the results obtained and the discussions. However, in the paper the authors do not clearly state whether the damage level is high or low, acceptable or not.
Thank you for the critical review. We have mentioned in the current version on the acceptability of the damage (Page 7; Line 272-73).
Also, I feel that the discussion would benefit from providing some terms of comparison for the level of damage corresponding to traditional operations (chainsaw and skidder that have been applied in the past in the studied stands). Otherwise, the readers are informed about the damage level derived from mechanized operations but are left wondering how the new mechanized technology compares with traditional logging technology;
The reviewer brings in a good point. We have provided general comparison of the results from this study with other mechanized operation (Page 9, Line 302-03; Page 10 Line 330-31; Page 11 Line 348- 50; Page 13 Line 362-63; Page 13 Line 386-89; Page 13 Line 402-403;)
Even though the study has not directly compared this mechanized version to that of semi-mechanized operation (traditional), studies such as Cudzik et al. (2017) has been mentioned and compared (Page 13 Line 414-20). We have again provided strategies on reduction of the residual stand damage which can be applied to both mechanized and semi-mechanized operations. On a general note, the objective of the study was to compare stand damage in different silvicultural prescriptions.
2 – It is not enough to take into consideration only the area (the size) of the wound in order to establish its severity (see the suggestions made in the text, line 223); In addition to size, other characteristics of the wound should be taken into account too, namely: the depth of the wound, the width of the wound relative to the circumference of the stem, the shape of the wound, etc.
The authors concur with the reviewer that area of the wound is not the only factor to establish severity. In the current study, along with wound size, visual observations were also used; which has been explained and specified in Page 6 Line 228-233. Figure 2 also visually explains this notion. The current index was developed based on similar visual observations made for other residual stand damage studies done in the region (Heisl 2013), which has been specified in the current version. The intention for specifying the wound size was for consistency and replication for future research.
3 – There are some inaccuracies in the data regarding the number of damaged trees (lines 280-289) that need to be corrected. It is recommended to check the data in the text and tables (for example Table 6);
Thank you for bringing this to our attention. We have corrected the number from 243 to 240 and 147 to 148 (Page 8 Line 289 and 293). Additionally, “Table 3” has been replaced to minimize ambiguity (Page 8 Line 291). We have re-checked the entire values throughout the document with our field data.
4 - In section - 3.3. Strategies adopted for minimizing residual stand damage - this part represents a general strategy, based on the studied bibliography. It would be interesting to come up with a strategy (for minimizing residual stand damage) developed on the basis of the results obtained and applicable to the stands studied under the given conditions and which could be reproduced in other stands located under similar conditions.
We believe Page 17 Line 489-506 describes the strategies that can be exclusively adopted in the region and the general challenges that occur during adoption within the region

Reviewer 3 Report
They are described in the file

Author Response
Response to Review for “Residual Stand Damage Under Different Harvesting Methods and Mitigation Strategies”
Dated June 7th 2021
Reviewer III
Residual Tree Damage
Add a more managerial explanation of the statistical model.
Managerial explanation such as how the statistical significance pertain to the field conditions has been explained through out the article. This concept has also been connected to other studies. A few examples are: Page 11 Line 348-50; Page 12 Line 356; Page 13 Line 386-389 Line 409-415. In rest of the cases significant differences were used to showcase the variation between the treatment or study sites.
Also…when however used in compound sentences needs a semicolon ie ( ; however, )
After going through the entire article we have put semicolon(;) prior to “however” in Page 15 Line 454.
Are the studies of decay relevant – may want to look at the resistance of wood for the species in the stand. Hemlock is known to be one of the worst. I find James Kiser studies in Douglas-fir taking a longer-term view of the problem of thinning damage, it is its value of the logs being produced… IE. Is the scare going to be bucked out of the final log…what is the change in value. I find a better approach to this problem than many others. Suggest you have a look at it. For the species can you discuss their durability or rot resistance at least in qualitative terms
After briefly going over the thesis, we came to see that the focus of the Kiser 2009 was on fungal infection. Even though the study is very interesting, it had little connection with the topic dealt in this article, ie. Residual stand damage due to varying silvicultural prescription. However, the authors would like to thank the reviewer for suggesting this work.
Can you justify the use of a probability proportion to size as a method to characterize the inventory. Why not fixed plots – are bigger trees more likely to be damaged than small. I would like to look at the volume, Trees per area by species type two. It is almost impossible to have perfect replicas for these types of study, but these may be covariate or factors that are later used to group the results to better explain the differences.
Probability proportional to size (PPS or variable radius plots) have an inherent disadvantage of sampling larger trees. This would be more suitable for commercial inventory; however, in this study Probability proportional to Frequency would provide with more representative data for the study area (by capturing all trees irrespective of their size).
The reviewer brings in an interesting concept of calculating the damage in terms of tree volume; However, this technique was not adopted for this study, with tree volume not being calculated. We have incorporated the volume concepts in terms of DBH. The concepts generally revolves around smaller trees likely to being damaged more than bigger trees; which was also evident in the current results. Similar results from other have been incorporated in the article (Page 15 Line 455-66).
The authors agree with the reviewer’s comment of not having exact replications; however an attempt was made to capture the variation within the treatment by having multiple transects at uniform distance. This has been further explained in Page 5 Line 208-07.
Can you justify that the systematic transects were more accurate. You have problem that the damage is concentrated near the major skid trails… it is not even throughout the stand. How does this overcome this problem?
The efficiency of systematic transects has been explained in Han 1997; where the researcher compared four different sampling techniques (systematic plot, random plot, blocks along skid trail, and systematic transect) and concluded systematic transect to be most efficient in terms of data qaulity. In this study systematic sampling suited the stand conditions and layout. This has been explained in Page 5 Line 205-13.
If you assume the damage is rot and limited to the area of damage – can you determine a value loss based on a bucking rules ie feasible log lengths… it would be interesting to look at more realistic solution.
The damaged trees were standing and data (scaling) on the defective log length were not collected, as this was not within the scope of the study. Therefore, this analysis could not be carried out. However, the authors recognize this to be a potential topic for future studies.
Furthermore, merchantable log length in the Northeastern US is subjected to market conditions and changes by species and DBH class. Therefore, this measurement could not be standardized, providing little practical information to the stakeholders. On another note, the intensity of merchantable trees removed varied between the silvicultural prescriptions, hence a direct comparison was not possible.
Additionally, data collection for the residual stand was completed within two months post-harvest. Therefore, fungal infection due to operational injury were not a part of the study design.
What would CANOVA look like with basal areas, volume removed, tree removed in the results? It would be interesting to see how these other variables influence the explanation of the variables.
The CANOVA test could not be conducted as the volume removed from each transect was not tracked. All the wood was brought into multiple landing and was not differentiated. Additionally, the pre- and post- harvest inventory plots were not aligned with the transects; therefore these parameters could not be further tested.
Damage types
Bumper trees have been suggested in the Pacific Northwest for years – they take the abuse and remove at the end of harvesting – protection the crop trees.
Thank you for the comment. That is correct. This has been mentioned in Page 13 Line 370-76.
The authors would like to thank the reviewer for the positive comments and critical review. This has helped us in significantly improving the quality and readability of the article.

Reviewer 4 Report
General comments
The paper has a high value for the forestry scientific community. In detail, I do not find any particular critical issues, but only minimal revisions. However, I have some doubts about how the paper is presented and its conclusions if the goal is “Sustainability” as a journal. I suggest to the authors to prepare an introduction that can better link to the sustainability (for example, damaged wood means wood that will not be destined to the sawmill industry and therefore will not be able to perform the function of fixing carbon dioxide for a long time). The conclusions should also take up the meaning of the paper in terms of sustainability.
Abstract and keywords
The abstract is enough clear. Anyway there is not the need to highlight in the specific the aims of the work. The first aim "evaluation of residual stand damage under different harvesting methods and silvicultural prescriptions" is enough general to involve also the following two "evaluate the intensity and frequency of damage " and "document strategies adopted globally".
The keywords are representative.
Introduction
L68 “A modified version of CTL, “hybrid 68 cut-to-length” (Hyb CTL)” it is clear the description but the reference are not enough to introduce this methods such as a “exhibits increased productivity” and “have less adverse impact on the regeneration”.
L96 the aims are clear enough. I have some doubts, reading them, if objectives 2 and 3 are also connected to the working method (WT and Hybrid CTL) and to the CTR, OSR, DLC prescription. This is not very clear and it should be clarified here.
Materials and Methods
Figure 1: It could be integrated with a map of the United States indicating the location of the sites. The map can be integrated and inserted in the margin of the main figure. It helps the reader to localize better the site
L210: It could be useful to clarify how the wounds of the trees located on the edge of the transepts were considered. This clarification is needed to replicate the study.
Again is missed the number of the Systematic transects and the methods on the choice of their locations.
The chapter 2.7 Literature review should specify the objective to which it is connected (discussion on "Strategies adopted for minimizing residual stand damage")
Results and Discussion
Table 1: not clear what is the meaning of “Amount of wood harvested (ton)” as well Skid trail (are the total length inside the block and average length for each skid trails? Please clarify it on the material and methods part related to the data collection.
The chapter is complete and it correctly reports the results of the work and the comparison with previous studies and practical experiences.
Author Response
Response to Review for “Residual Stand Damage Under Different Harvesting Methods and Mitigation Strategies”
Dated June 7th 2021
Reviewer IV
General comments
The paper has a high value for the forestry scientific community. In detail, I do not find any particular critical issues, but only minimal revisions. However, I have some doubts about how the paper is presented and its conclusions if the goal is “Sustainability” as a journal. I suggest to the authors to prepare an introduction that can better link to the sustainability (for example, damaged wood means wood that will not be destined to the sawmill industry and therefore will not be able to perform the function of fixing carbon dioxide for a long time). The conclusions should also take up the meaning of the paper in terms of sustainability.
The authors would like to thank the reviewer for this comments. One sentence (Page 2 Line52-54) has been added in the opening paragraph to establish ties with the scope of the journal and article. In addition, there is also a sentence reffering to sustainable forest management in the conclusion (Page 18 Line 528-529)
Abstract and keywords
The abstract is enough clear. Anyway there is not the need to highlight in the specific the aims of the work. The first aim "evaluation of residual stand damage under different harvesting methods and silvicultural prescriptions" is enough general to involve also the following two "evaluate the intensity and frequency of damage " and "document strategies adopted globally".
The authors would like to express their appreciation for the comments. The first objective is referring to the damages due to the prescription; whereas, the second is focusing on the type of damage (bole, root and canopy).
The keywords are representative.
Introduction
L68 “A modified version of CTL, “hybrid cut-to-length” (Hyb CTL)” it is clear the description but the reference are not enough to introduce this methods such as a “exhibits increased productivity” and “have less adverse impact on the regeneration”.
Hybrid cut-to-length is a harvesting system very unique to Northeastern America (Canada and USA). There has been no studies conducted (up to date and our knowledge) on the Hybrid cut-to-length method and its impact on the residual stand. The only available study on Hyb CTL is on the productivity and cost of operations by Louis and Kizha 2021. However, it is widely accepted with in the industry that such operations tend to exhibit better productivity and reduced stand damage compared to the traditional cut-to-length. In order to address this comment, we have incorporated “ is widely believed to” in Page 2 Line 73-74.
L96 the aims are clear enough. I have some doubts, reading them, if objectives 2 and 3 are also connected to the working method (WT and Hybrid CTL) and to the CTR, OSR, DLC prescription. This is not very clear and it should be clarified here.
The authors acknowledge the comment made by the reviewer. In this study, the first objective was to evaluate residual stand damage incurred due to three silvicultural prescriptions. Each study site (where the prescriptions were applied) had different harvesting methods (WT and Hyb CTL). We have accounted for the damage due to prescription and harvesting methods separately in the sections, “3.2.2. Impacts due to harvesting methods and silvicultural prescriptions”. The difference between treatments were also compared in table 6.
Materials and Methods
Figure 1: It could be integrated with a map of the United States indicating the location of the sites. The map can be integrated and inserted in the margin of the main figure. It helps the reader to localize better the site.
The authors acknowledge the comment. Attempts were made to portray the study sites in a map of the USA; however, at this scale the study sites could not be distinguishable, which was why the state map was chosen instead. Additionally, the latitude and longitude data was provided for each of the study area.
L210: It could be useful to clarify how the wounds of the trees located on the edge of the transepts were considered. This clarification is needed to replicate the study.
The trees that were only within the transect were counted. This was ensured by physically measuring the distance to the center of the trees along the vertical axis from the center of the transect and using real-time GPS location (Avenza Map). Trees which fell outside the transect boundary were excluded from the study. This has been incorporated in the current version Page 5 Line 213-215. The GPS location for all trees were recorded, which help in identifying the trees for future.
Again is missed the number of the Systematic transects and the methods on the choice of their locations.
The authors would like to thank the reviewer for bringing this up. Sufficient information has been added to the current version specifying the number of transect and layout details (Page 5 Line 205-08).
The chapter 2.7 Literature review should specify the objective to which it is connected (discussion on "Strategies adopted for minimizing residual stand damage")
This has been taken care in the current version. Please see Page 6 Line 250.
Results and Discussion
Table 1: not clear what is the meaning of “Amount of wood harvested (ton)” as well Skid trail (are the total length inside the block and average length for each skid trails? Please clarify it on the material and methods part related to the data collection.
The authors are not sure on this comment. The amount of wood harvested refers to the merchantable wood generated during respective harvest for each treatment. For better clarification this has been changed to “total wood harvested” and “skid trail length” Page 7 Table 1.
The chapter is complete and it correctly reports the results of the work and the comparison with previous studies and practical experiences.
The authors would like to express their gratitude for the constructive comments.
The authors would like to thank the reviewer for the positive comments and critical review. This has helped us in significantly improving the quality and readability of the article.

Round 2
Reviewer 3 Report
Still using or never tested other statistical models but ANOVA.